# Learning to Anneal and Prune Proximity Graphs for Similarity Search

## Abstract

This paper studies similarity search, which is a crucial enabler of many feature vector–based applications. The problem of similarity search has been extensively studied in the machine learning community. Recent advances of proximity graphs have achieved outstanding performance through exploiting the navigability of the underlying graph structure. In this work, we introduce the annealable proximity graph (APG) method to learn and reshape proximity graphs for efficiency and effective similarity search. APG makes proximity graph edges annealable, which can be effectively trained with a stochastic optimization algorithm. APG identifies important edges that best preserve graph navigability and prune inferior edges without drastically changing graph properties. Experimental results show that APG achieves state-of-the-art results not only by producing proximity graphs with less number of edges but also speeding up the search time by 20–40% across different datasets with almost no loss of accuracy.

## 1 Introduction

Similarity search (nearest neighbor search) is an integral and indispensable task in many machine learning applications, such as non-parametric classification/regression, computer vision, information retrieval, and language modeling. Recently, it has been demonstrated that it is possible to build a *vector search engine* to support *semantic search* (Chen et al., 2018; Sullivan, 2018; Wang et al., 2018a; Johnson et al., 2017), which leverages high-quality neural ranking models (Nogueira & Cho, 2019; Xiong et al., 2017; Zamani et al., 2018a) to encode both natural language query and documents into dense continuous feature vectors and performs similarity search to retrieve relevant documents with vast data volumes (e.g., based on Euclidean distance). This approach has demonstrated significant relevance gains in a wide range of applications and outperforms existing term matching baselines, such as web search (Huang et al., 2013; Zamani et al., 2018b), question and answering (Yu et al., 2014), ad-hoc retrieval (Mitra et al., 2017; Dehghani et al., 2017; Guo et al., 2016), mobile search (Aliannejadi et al., 2018), and product search (Van Gysel et al., 2016).

The efficiency and effectiveness of the similarity search approaches have become a problem of great interest, due to the widespread commercial value and the exciting prospect. Recent advance of *proximity graphs* has demonstrated great potential for fast and accurate nearest neighbor retrieval (Malkov & Yashunin, 2016; Fu et al., 2019), and the empirical performance of proximity graphs outperforms existing tree–based (Bentley, 1975; Beckmann et al., 1990; Yianilos, 1993; Muja & Lowe, 2014), locality sensitive hashing–based (Gionis et al., 1999), and product quantization–based methods (Jegou et al., 2011; Ge et al., 2013; Norouzi & Fleet, 2013; Lempitsky, 2012; Kalantidis & Avrithis, 2014) by a large margin. Proximity graphs exploit the navigability of graph structures, which the search process relies on to converge and achieve good efficiency. In practice, that often results in dense connectivity and large memory consumption because they need to have sufficient edges to maintain specific graph properties, which is a major limitation of this class of approaches.

We wish to improve the efficiency of similarity search. In this paper, we address the following research question: can we learn to prune edges of a proximity graph while still being accurate to find nearest neighbors? Specifically, the pruned proximity graph should be more efficient than the state-of-the-art proximity graphs with comparable accuracy. Before providing a definite answer to the question, we briefly review the findings in percolation theory that motivates our research.

Percolation describes the phase transition of a physical system when one or more of its properties change abruptly after a slight change in controlling variables (e.g., temperature, pressure, or others) (Broadbent & Hammersley, 1957). Prototypical percolation processes include water turning into ice or steam, and the spontaneous emergence of magnetization and superconductivity in metals. Percolation theory mathematically models these physical systems as complex networks and phase transition as a dramatic change of the properties of network connections. We believe that if we can model edge importance as the robustness of proximity graphs to the removal of the edges between vertices, we can produce a proximity graph with less number of edges without dramatically changing the navigability of the graph.

We present **A**nnealable **P**roximity **G**raph (APG), for simplifying proximity graphs. In particular, we make the following contributions:

- We introduce the annealable proximity graph and summarize its key characteristics.
- To learn edge importance, we present a percolation inspired method for identifying important edges and introduce a domain-specific loss derived from search distance errors.
- Our formulation makes it possible to leverage a stochastic optimization algorithm to optimize the objective and prune edges with low importance.
- We prove the convergence of our optimization process and a theoretical guarantee of the search quality of the pruned graph.

This approach is unique compared with previous proximity graph algorithms, where most of them only exploit the structure of the underlying index instead of learning from query distribution to reshape proximity graphs. We provide a detailed empirical analysis of our approach. Experimental results show that our approach reduces the number of edges of state-of-the-art proximity graphs significantly by 50% while also speeds up the search time by 20–40% across different datasets with almost no loss of accuracy.

## 2 RELATED WORK

In this section, we review the main ideas from the existing work that is relevant to our approach.

**Approximate nearest neighbor search (ANN).** The problem of similarity search has been extensively studied in the literature of ANN algorithms, which trade the guarantee of exactness against high-efficiency improvement. Some representative methods include tree structure–based (Bentley, 1975; Beckmann et al., 1990; Yianilos, 1993; Muja & Lowe, 2014), locality sensitive hashing (LSH)–based (Gionis et al., 1999), product quantization (PQ)–based (Jegou et al., 2011; Ge et al., 2013; Norouzi & Fleet, 2013; Lempitsky, 2012; Kalantidis & Avrithis, 2014), and nearest neighbor graph–based (Hajebi et al., 2011; Fu & Cai, 2016) approaches. Although some of these methods, such as LSH, have strong theoretical performance guarantee even in the worst case (Indyk & Motwani, 1998), recent advances of the *proximity graphs* have demonstrated logarithmic search complexity and outperformed prior approaches by a large margin (Malkov & Yashunin, 2016; Douze et al., 2018; Fu et al., 2019; Li et al., 2019).

**Proximity graphs.** A proximity graph exploits the closeness relationship among feature vectors to support similarity search. In particular, let $V = \{v_i \in \mathbb{R}^D | i = 1, ..., N\}$ be a database of vectors, a proximity graph $G(V, E)$ is a directed graph, where each vertex corresponds to one of the vectors $v$ and the whole graph achieves great local connectivity (as in a lattice graph) combined with a small graph diameter (as in a random graph) (Malkov et al., 2014; Malkov & Yashunin, 2016; Fu et al., 2019). Such a graph exhibits strong navigability and enables quick search with an $N$-greedy best-first search algorithm. During the search, a candidate queue of size $L$ is used to determine the trade-off between the searching time and accuracy. Recent studies look into optimizing proximity graphs with product quantization (Douze et al., 2018; Baranchuk et al., 2018). However, these approaches often suffer from a considerable amount of recall loss on large datasets because quantization errors tend to be large on dense continuous feature vectors (e.g., generated by neural networks).

**Learning to prune.** Pruning is a common method to derive sparse neural networks and reduce their heavy inference cost (Han et al., 2015a;b). These methods annihilate the non-important weights

through the introduction of an $L_0$ or $L_1$ regularizer to the loss function. Many of them remove weights and keep important weights to best preserve the accuracy. Neural network pruning can also be viewed as an architecture search technique, where the network is viewed as a computational graph where the vertices denote the computation nodes and the edges represent the flow of tensors (Wang et al., 2018b; Liu et al., 2019). To the best of our knowledge, learning to prune has not yet been applied to the task of proximity graphs for similarity search. However, it appears to be a natural fit for restructuring the proximity graphs to improve similarity search.

## 3 CHALLENGES

The navigability of proximity graphs comes as a result of approximating monotonicity graphs, e.g., Delaunay graphs (Lee & Schachter, 1980). According to the graph monotonicity theory (Dearholt et al., 1988), a monotonicity graph has a strong guarantee to find the exact nearest neighbor by following a *monotonic path* with 1-greedy search (Fu et al., 2019). However, monotonicity graphs in high dimensional space quickly become almost fully connected, and search in fully connected graphs would be infeasible, due to the *out-degree explosion problem* (Hajebi et al., 2011). To address the problem, proximity graphs limit each node to connect to only a number of $R$ neighbors, aiming to minimize the loss of graph monotonicity while still letting greedy search be effective.

However, several challenges remain. The correct choice of $R$ is not so obvious. $R$ cannot be too small, because then the graph tends to lose too much monotonicity and search can frequently get stuck at non-global local minima, hurting accuracy. A sufficiently large $R$ is often required to reach high accuracy, but it also increases the number of edges significantly and decreases efficiency: (1) It ubiquitously raises the connections of both "hubs" (i.e., nodes in dense areas) and nodes in sparse areas; and (2) it makes the graph more densely connected with many vertices sharing a lot of common neighbors, increasing unnecessary distance computations. Ideally, each vertex should have a different R that best preserves graph monotonicity. The problem is beyond selecting a good value for $R$ and seems to require more fundamental changes to existing proximity graphs.

## 4 METHODS

In this section, we propose the annealable proximity graph (APG), which supports the exploration of edge heterogeneity in proximity graphs and learning to prune such graphs with an eye towards getting to the nearest neighbor as quickly as possible with minimal loss of accuracy.

### 4.1 OVERVIEW

APG starts with augmenting a pre-built proximity graph with a learnable weight associated with each edge, representing its importance to preserve graph monotonicity, and a keep probability (§§ 4.2). The weight is updated in an iterative manner according to a learning rule.

APG employs a multi-step learning process, as illustrated in Fig. 1. At initialization of APG, all edges have equal weights. Since no edge heterogeneity has been learned so far, applying pruning at this stage could lead to premature decisions. Therefore, we start with performing a "warm-up" iteration over the edge weights only, without optimizations, i.e., as the grace-period.

Once this warm-up ends, we introduce a systematic and principled approach to optimize the *edge keep probability distribution* of the APG (§§ 4.4). In particular, APG models edge importance as the robustness of graph monotonicity to edge removal and defines an objective function that reflects the destruction of graph monotonicity, based on relative search distance errors (§§ 4.3). It then generates a sequence of randomized subgraphs through a sampling policy to learn edge importance and uses a predefined annealing schedule to optimize the objective function.

The process ends once we meet a stopping criterion. After this step, APG marks low weight edges as less important and perform a hard pruning to remove inferior edges, as shown in Fig. 2.

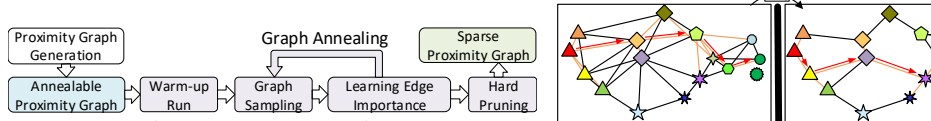

Figure 1: APG overview.

Figure 2: Example of before and after pruning proximity graph.

## 4.2 ANNEALABLE PROXIMITY GRAPH (APG)

Given a proximity graph $G(V, E)$, an annealable proximity graph $G^*(V, E)$ is obtained by augmenting each edge $e \in E$ with a weight variable $w_e$, and a *keep probability function* $p : E \to (0, 1)$ such that $p(e) \equiv p_e$ indicates the keep probability of $e \in E$. That is, an independent Bernoulli random variable $R_e$, where $\mathbb{P}(R_e = 1) = p_e, \mathbb{P}(R_e = 0) = 1 - p_e$ is assigned to each $e$.

Intuitively, $p_e$ should be: (i) monotonically increasing as $w_e$ increases; and (ii) $\lim_{w_e \to +\infty} p_e = 1$, and $\lim_{w_e \to -\infty} p_e = 0$. More importantly, it is desirable to have $w_e$ initialized to similar values for all edges, allowing each edge to have an equal probability of consideration when there is little information about edge importance. As the optimization process continues, $p_e$ should converge into a degenerated distribution that allows identifying a subset of removable edges that do not significantly change graph properties. Moreover, we introduce an additional parameter, the temperature $T \in (0, \infty)$, which smooths the probabilities $p_e$ as following. If $T \to \infty$ (at the beginning), the probabilities $p_e$ converges uniformly to the same value regardless of edge $e$; on the other hand, if $T \to 0$ (at the end), the probabilities $p_e$ converges to either 1 or 0, for important and not important edges, respectively. To satisfy the above conditions, we introduce the following function $p$:

$$p_e(T) = \frac{1}{1 + \exp\left(-\frac{w_e + \mu}{T}\right)} \tag{1}$$

where $\mu$ is a normalizing factor to keep $\sum_{e \in E} p_e(T) = C$ a constant.

## 4.3 ROBUSTNESS OF GRAPH MONOTONICITY

To efficiently find nearest neighbors, all the previous algorithms try to exploit the proximity graph structure of $V$ vectors by letting each vertex connect a number of $R$ neighbors. However, some connections could be more crucial for preserving graph monotonicity, which is important for search efficiency, while the rest is less important. How do we identify those important edges?

One naive approach is to keep those edges that appear as part of the shortest path from the entry vertex to the ground truth nearest neighbor for a query. Such an approach significantly suffers overfitting: non-shortest-path edges may still contain possibly relevant closeness relation for unseen queries on the test query set. Another possibility is to treat all checked edges during the search process as important. However, the checked and taken edges are then not differentiated, and some checked but not taken edges may even hurt accuracy by misleading the route. Both of these results are undesirable. As § 3 mentioned, proximity graphs rely on the approximation of graph monotonicity to converge and achieve their efficiency. Can we identify important edges based on their robustness to preserve graph monotonicity?

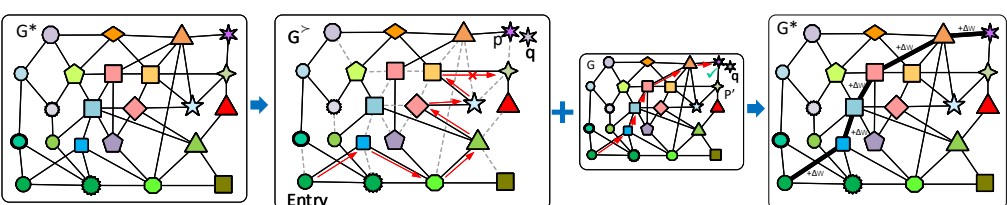

Figure 3: A high-level depiction of learning edge importance for proximity graphs.

**Identifying important edges.** Inspired by the phase transition in *percolation* (Broadbent & Hammersley, 1957; Callaway et al., 2000), we introduce a method for identifying edge importance in proximity graphs. In particular, as Fig. 3 shown, for a given $G^*$, we randomly delete each edge $e$ with probability $1 - p_e$ (remember that $p_e$ denotes the keep probability of $e$), independently of all other edges, and we denote the resulting random graph by $G^{\succ}(V, E \setminus F)$, where $F$ is the set of deleted edges. For any query $q$, if we can find the exact nearest neighbor in $G$ but not in $G^{\succ}$ under a search budget, then we treat the *edge hops* along the search path of $q$ in $G$, including those erroneous paths, to be important for preserving the robustness of graph monotonicity, because it is their deletion that causes the failure to find the nearest neighbor in $G^{\succ}$.

**Relative distance errors.** Once we get the set of edge hops for a query $q$ (let us denote it as $H^q$), we update the weight $w_h$ of $h \in H^q$ to increase the importance of those edges, based on how far off the found candidate is relative to the true nearest neighbor, similar to the Teacher Forcing scheme (Williams & Zipser, 1989). In particular, assume $G$ answers the query $q$ by returning a point $p$, and $G^{\succ}$ answers $q$ with a found candidate $p'$, we define the weight update as:

$$\Delta w_h = (\frac{\delta\langle p', q \rangle}{\delta\langle p, q \rangle} - 1) \cdot \eta, \forall h \in H^q \tag{2}$$

where $\delta\langle \cdot, \cdot \rangle$ represents the distance between two vertices and $\eta$ is the learning rate. It measures the delta of searching $q$'s nearest neighbor in a subgraph $G^{\succ}$ versus in the full graph $G$ by removing $F$ edges. The update is designed based on the following intuition: when the returned candidate $p'$ in $G^{\succ}$ is the exact nearest neighbor as $q$ returned by $G$, then the deleted edges are less important and the relative distance error is 0. Otherwise, the larger $\delta\langle p', q \rangle$ in comparison to $\delta\langle p, q \rangle$, the more importance $H^q$ indicates, and the edge weights of $H^q$ should increase more.

**Objective.** Based on Eqn. 2, we propose to learn the keep probability distribution of APG while minimizing the relative distance error over a learning set $Q$. In particular, the learning objective function is defined as:

$$\text{minimize} \quad \frac{1}{|Q|} \sum_{q \in Q} \mathbb{E} \left[ \sum_{h \in H^q} \Delta w_h \right] \tag{3}$$

Based on the learned keep probability, we identify a subgraph $G^{\succ}$ that is robust to the deletion of a subset of edges $F$ with minimal loss of graph monotonicity.

## 4.4 OPTIMIZATION

The main objective of the optimization is to construct a proper learning framework to optimize the objective defined in Eq 3. Similar to bagged ensemble learning (Bengio et al., 2017), one approach is to initialize a set of graphs, each with a subset of edges randomly deleted. Then it individually learns edge importance on each subgraph and finally combines learned weights. However, bagging ignores the dependency among the individual subgraphs, and the edge importance does in fact depend on the chosen subgraph. In this paper, we introduce a learning algorithm, inspired by simulated annealing (Ingber, 1993) and stochastic optimization process (Bottou & Bousquet, 2007), where we probabilistically generate a sequence of sampled subgraphs to learn the important edges that preserve graph monotonicity. The benefit is that it gradually discovers important edges while also allowing edges with lower keep probability to be sampled to demonstrate their values.

**Generating a sequence of random subgraphs.** For a given $G^*$, we generate a sequence of randomized subgraphs: $G^{\succ}(1) \to G^{\succ}(2) \to \cdots \to G^{\succ}(K)$, which correspond to $K$ optimization steps. At each step $k$, the keep probability of each edge is computed from the weight of the previous step $w(k-1)$. The subgraph $G^{\succ}(k)$ is obtained via sampling edges from $G^*$ by randomly picking a set of $E(k)(|E(k)| \leq |E|)$ edges based on this keep probability. The new weight $w(k)$ is then obtained through learning to minimize the relative distance errors measured on $G^{\succ}(k)$. As a stringent control, the expected number of edges to be selected by the algorithm is $|E(k)| = \lceil \lambda(k) \cdot |E| \rceil$, where $0 < \lambda(k) \leq 1$ denotes the sampling ratio at iteration $k$ and is governed by a sampling policy.

**Gradual sampling policy.** The sampling policy $\lambda(k)$ decides how many edges a sampled sub-graph $G^\succ(k)$ has. We found that the simplest solution to having a fixed value $\lambda(k) \equiv 1 - \sigma$ works reasonably well in most cases, where $\sigma$ is the final pruning ratio (the fraction of pruned edges). In this paper, inspired by the formula suggested in (Zhu & Gupta, 2018), we define $\lambda(k)$ as:

$$\lambda(k) = 1 - \sigma + (\lambda(0) + \sigma - 1)\left(1 - \frac{k}{K}\right)^c \tag{4}$$

where $c \in \{1, 3\}$ and $\lambda(0)$ is the initial sampling rate. Take $\sigma = 0.5$ and $\lambda(0) = 1$ as an example, intuitively, this policy allows to select more edges for exploration in the beginning and gradually becomes more selective as the optimization is close to the end.

**Binomial weight normalization.** The expectation of the number of edges of a random subgraph $G^\succ(k)$ follows a Poisson binomial distribution: It is the sum of Bernoulli random variables $R_e, e = 1, ..., |E|$, each taking on values 0 and 1 with probabilities $1 - p_e$ and $p_e$, respectively. However, since $\Delta w \geq 0$ in Eq 2, how to expect the sampled graph $G^\succ(k)$ to have $|E(k)|$ edges given that increased weights also increase the sum of $R_e$? To address this issue, we do a binomial normalization to adjust the edge weights at each iteration by adding a normalizing factor $\mu(k)$ (as in Eq 1) to all edge weights so that the sum of $R_e$ equals to $|E(k)|$:

$$|E(k)| \equiv \mathbb{E}\left[\sum_{e \in E} R_e\right] = \sum_{e \in E} p_e(T) = \sum_{e \in E} \frac{1}{1 + e^{-\frac{w_e + \mu(k)}{T}}} \tag{5}$$

where $\mu(k)$ is calculated through a binary search of the computed sum of probabilities, which has a time complexity of $O(|E| \cdot \log(\max(w(k)) - \min(w(k))))$.

**Annealing schedule.** To balance exploration and exploitation of edge importance, our approach includes an annealing schedule $\Phi(k)$ to determine the temperature $T$, which is updated along the iterations. The schedule choice is dominated by a trade-off. On one hand, fast temperature decay simplifies the optimization objective and reduces the complexity, assisting the edge weights to converge, since inferior edges quickly have their keep probability driven to 0 and are excluded from subsequent optimizations. However, premature decisions could lead to a sub-optimal keep probability distribution. This suggests that we should choose a slow temperature decay, which provides more opportunities for testing random subgraphs during the pre-convergence phase, but also may find a better subgraph during the convergence phase.

The solution we propose is based on observations explored in simulated annealing, which share a similar trade-off between the continuity of the optimization process and the time required for deriving the solution (Ingber, 1993). Among the many alternatives, we choose the exponential schedule, which has been shown to be effective in multiple other tasks, e.g. (Kirkpatrick et al., 1983; Nourani & Andresen, 1998),

$$\Phi(k) = T_0 \cdot \beta^k \tag{6}$$

This schedule starts with a relatively high temperature $T_0$ and decays fast with a decay factor $\beta$.

**Hard pruning.** The optimization process ends when it meets a stopping criterion. In our current implementation, we use a simple criterion that stops after a given number of iterations. Once the optimization finishes, we re-rank edge weight and prune less important edges according to a desired pruning ratio $\sigma$. While pruning, we avoid deleting bridges that disconnect the graph. After pruning, we add a minimal number of edges to keep the graph strongly connected through connectivity augmentation (Hsu et al., 2017). The overall learning algorithm is given in Algorithm 1.

**Convergence and correctness proof.** We demonstrate the effectiveness of our approach empirically in § 5 and provide a theoretically derived proof of the convergence and correctness of our algorithm in Theorem A.1 (Appendix A).

---

**Algorithm 1**                                                             **APG learning algorithm**

---

1: **Input:** Unpruned proximity graph $G(V, E)$, learning set $Q$, candidate queue length $L$.
2: **Output:** Pruned graph $G(V, E \setminus F)$.
3: **Parameters:** Learning rate $\eta$, starting temperature $T_0$, decay factor $\beta$, pruning ratio $\sigma$, max iteration $K$
4: **Init:** $T \leftarrow T_0, k \leftarrow 0$
5: Convert $G(V, E)$ to $G^*(V, E)$, $w \leftarrow 0$
6: Update $w$ according to a warm up run
7: **while** $k \leq K$ **do**
8:     $\lambda \leftarrow 1 - \sigma + (\lambda(0) + \sigma - 1)\left(1 - \frac{k}{K}\right)^c$
9:     Normalize $w$ s.t. $|E(k)| = \lceil \lambda(k) \cdot |E| \rceil \equiv \sum\limits_{e \in E} \frac{1}{1 + \exp\left(-\frac{w_e + \mu(k)}{T}\right)}$
10:     Randomly sample a subgraph $G^\succ(k)(\mathbf{V}, E(k))$
11:     **for** $q$ **in** $Q$ **do**
12:         $p', \leftarrow search(G^\succ(k), q, L)$
13:         $p, H^q \leftarrow search(G, q, L)$
14:         **if** $p \neq p'$ **then**
15:             **for** $h$ **in** $H^q$ **do**
16:                 $\Delta w_h \leftarrow \left(\frac{\delta\langle p', q\rangle}{\delta\langle p, q\rangle} - 1\right) \cdot \eta$
17:                 $w_h \leftarrow w_h + \Delta w_h$
18:     $T \leftarrow T_0 \cdot \beta^k$
19:     Shuffle $Q$
20: Remove $|F| = \sigma \cdot |E|$ lowest ranking edges from $G^*$
21: Convert $G^*(V, E \setminus F)$ to $G(V, E \setminus F)$

---

## 5   EVALUATION

### 5.1   METHODOLOGY

**Datasets.** We evaluate APG on three publicly available datasets, which are widely used as the similarity search benchmarks:

- **SIFT1M** is a classical dataset containing 128-dimensional SIFT descriptors (Jegou et al., 2011). It consists 1,000,000 base vectors, 100,000 learning vectors, and 10,000 testing vectors.

- **Deep1M** is a random 1,000,000 subset of one billion of 96-dimensional vectors produced by CNN (Babenko & Lempitsky, 2016). We sample 100,000 vectors from the provided 350M learn set as the learning set. For testing, we take the original 10,000 queries.

- **GloVe** is a collection of 200-dimensional word embedding vectors from Twitter data (Pennington et al., 2014). We randomly sample from the original 1,193,514 vectors to get base, learning, and testing sets, each containing 1,000,000, 100,000, and 10,000 vectors, respectively.

For each dataset, we use one of the state-of-the-art approaches, Hierarchical Navigable Small World graph (Malkov & Yashunin, 2016), to build the proximity graph using the base set, which we refer to as PG (proximity graph). We use the learning set to learn and prune the bottom layer of HNSW and use the testing set for final evaluation. We do not prune the edges in the upper layers of HNSW since there are much fewer edges in those layers. We set hyperparameters as $T_0 = 1$, $K = 20$, $\beta = 0.8$, $\eta = 0.1$, and $\sigma = 0.5$. The training time is 77, 40, and 128 minutes for SIFT1M, Deep1M, and GloVe, respectively.

**Setup.** All the experiments were done on a 64-bit Linux Ubuntu 16.04 server with Intel Xeon CPU E5-2650 v4 @ 2.20GHz processor.

**Implementations.** APG and the learning algorithm are implemented in C++. Subgraph sampling is implemented by using a binary mask map, which is of the same size as the number of edges, to determines edges that are kept in the sampled subgraph. The weights that are masked in the sampled subgraph do not get updated during the weight update phase.

**Evaluation metrics.** To measure the pruning effectiveness, we report the number of edges before and after pruning and the search time. We measure the accuracy by calculating the rate of queries for which the exact nearest neighbor is found. Since it is essential to be both fast and with high accuracy, we focus on the high accuracy range.

## 5.2 EXPERIMENT RESULTS

In this section we evaluate the proposed method by comparing the following schemes for each dataset:

- **Original PG.** We construct a proximity graph over the base vectors using the HNSW algorithm.

- **Ours.** Our main algorithm as described in § 4. We use the same initial $R$ as PG and set the pruning ratio to $\sigma = 0.5$, which forces half of the edges to be useful for preserving the graph properties.

- **PG + sampling.** As a heuristic baseline, we uniformly sample half of the edges from PG to remove without iteration optimizations and annealing edge weights.

- **Sparse-PG.** Unlike ours, this approach directly constructs a proximity graph with only half edges to begin with. Hence all the edges still have equal importance.

Table 1 shows the accuracy, the edge counts, the search time for all three datasets. To compare the performance of baselines and our approach apples to apples, we perform controlled experiments to keep all approaches to reach the same accuracy target in order to compare the edge reduction rate and the search time improvement. Given an accuracy target (e.g., 0.99 for SIFT1M, 0.94 for Deep1M, and 0.83 for GloVE), we vary $L$ to find the minimum latency to reach the desired accuracy.

Overall, compared to the original proximity graph, ours reduces the number of edges by 50%, making PG more memory efficient. On SIFT1M and GloVe1M, ours further makes the search 26.1% and 1.8% faster, respectively. This is because pruning simplifies the proximity graph while preserving the graph monotonicity, so that a query still gets to the nearest neighbor but is faster by avoiding checking less important edges. On Deep1M, the fastest search time is achieved by the unmodified PG, because when the target accuracy is really high (e.g., 0.99), there are much fewer redundant edges. And for this particular dataset, after pruning 50% edges, ours needs to search a little bit more to reach the same level of accuracy.

Both *PG + subsampling* and *sparse-PG* have a similar number of edges like ours. However, reducing the number of edges alone does not necessarily lead to better search efficiency. If the search takes a fixed number of hops to reach the nearest neighbor, then pruning edges will lead to less number of vectors being checked. However, for proximity graphs, the search stop condition is (1) greedy search until reaching a local optimum and (2) the search time budget has exhausted, so the search may end up searching the same amount of edges even after pruning edges. Since subsampling does not take into graph monotonicity into account, it leads to poor search efficiency either because a query needs to take a detour to find the nearest neighbor or it may not even find the nearest neighbor if the graph becomes disconnected. In fact, randomly removing edges is so destructive that GloVe1M cannot get the desired 0.83 accuracy even if the search time has increased from 0.55ms to 1.45ms. On the other hand, the sparse-PG directly restricts the number of edges of each node during the graph construction phase. However, as described in § 3, a smaller $R$ hurts the connectivity of the proximity graph, and search can easily get stuck at local minimum. In order to reach the same accuracy, this approach needs to search more, which causes the degradation of search time. As a result, APG makes the search 25–74.2% faster compared to these two approaches.

**Impact of pruning ratio $\sigma$.** We evaluate the impact of different pruning ratio $\sigma$. Fig.4a shows that, as $\sigma$ increases, the number of edges left in the pruned graph linearly decreases, which is expected because our algorithm enables accurate control of pruning a given number of edges.

Fig.4b evaluates the impact on accuracy varying $\sigma$, under different $L = 20, 50, 100$. The accuracy remains almost on par as the unpruned proximity graph when the pruning ratio is equal or less than 50%. It starts to drop significantly once the pruning ratio is beyond 50%. The results suggest that our approach prunes a large number of edges while still being able to let queries find their nearest

| Dataset | Configuration | Recall@1 | #Edges | Latency(ms) | Edge reduction rate | Search time improvement |
|---------|---------------|----------|--------|-------------|---------------------|------------------------|
| SIFT1M | Original PG | 0.993 | 40.3M | 0.23 | 50.0% | 26.1% |
| | PG + sampling | 0.992 | 20.2M | 0.43 | 0.0% | 60.5% |
| | Sparse-PG | 0.991 | 20.2M | 0.66 | 0.0% | 74.2% |
| | Ours | 0.993 | 20.2M | 0.17 | | |
| Deep1M | Original PG | 0.947 | 43.2M | 0.1 | 50.0% | -4.0% |
| | PG + sampling | 0.945 | 21.6M | 0.16 | 0.0% | 35.0% |
| | Sparse-PG | 0.943 | 21.6M | 0.14 | 0.0% | 25.7% |
| | Ours | 0.943 | 21.6M | 0.104 | | |
| GloVe1M | Original PG | 0.831 | 20.6M | 0.55 | 50.0% | 1.8% |
| | PG + sampling | 0.814 | 10.3M | 1.45 | 0.0% | 62.8% |
| | Sparse-PG | 0.83 | 10.3M | 0.74 | 0.0% | 27.0% |
| | Ours | 0.833 | 10.3M | 0.54 | | |

Table 1: Comparison between APG and different baselines over SIFT1M, Deep1M, and GloVe.

neighbors, as long as the proximity graph has not entered a phase transition where it starts to quickly lose graph monotonicity.

To demonstrate the impact of the pruning ratio on search efficiency, we present Fig.4c and Fig.4d, which illustrate the distance computations and search time under different $\sigma$. The results show that larger $\sigma$ often leads to fewer distance computations and shorter search time, with varying $L$. This is expected, because APG makes the proximity graph sparser and saves both distance computations and search time by avoiding checking less important edges.

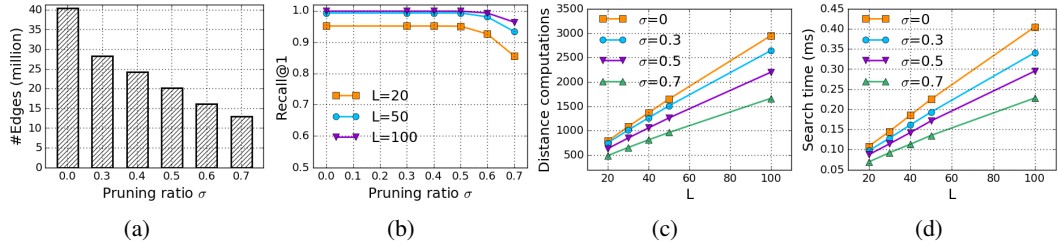

(a)      (b)      (c)      (d)

Figure 4: Impact of different pruning ratio. (a) #Edges v.s. pruning ratio; (b) Accuracy v.s. pruning ratio under different $L$; (3) and (4) Distance computations and search time under different $\sigma$.

**Impact on degree distribution.** To see how pruning affects the graph topology, Fig.6 in Appendix B shows the frequency distributions of in-degree and out-degree for all nodes before and after pruning ($\sigma = 0.5$). We found that the out-degree tends to be smoothed out to have a power-law distribution after the pruning, and the in-degree remains to have a binomial distribution but with only a slight shift after the pruning.

**Internal of APG learning process.** To reveal the internal learning process, Fig. 7 in the Appendix demonstrates the snapshot of the keep probability distribution at different iterations. The distribution starts with a relatively high temperature and high sampling ratio $\lambda$, so that most edges having an equal probability being selected (e.g., 0.8-1). Then as the iteration goes on, the temperature decays by following the annealing schedule $\Phi(k)$ (Eqn. 6) and the sampling ratio follows the sampling policy $\lambda(k)$ (Eqn. 4). As a result, those edges that are less important for preserving graph monotonicity have their probability gradually reduced. This process keeps going until the distribution becomes very biased: most of the edge probability are distributed around the two peaks. This is expected because as Theorem A.1 shows, the optimization eventually converge with the joint distribution of $R_e(T)$ for all edges $e$ being equal for $T \to \infty$ and sparsely supported for $T \to 0$.

## 5.3 COMPARISON WITH EXISTING METHODS

We include the comparison between APG and two state-of-the-art proximity graphs: (1) HNSW (Malkov & Yashunin, 2016) and (2) NSG (Navigable Spread-out Graph) (Fu et al., 2019).

We further report the comparison group results from the corresponding *sparse* counterparts: (3) HNSW-sparse and (4) NSG-sparse, both of which have a similar number of edges like ours. We also provide a comparison to (5) HNSW with random pruning [1].

We first consider the compromise between the search time and accuracy. Fig. 5 illustrates the accuracy–latency tradeoff between APG and the other configurations. We observe that on SIFT1M, APG is reaches the highest accuracy (0.995) when having the same search time budget (e.g., 0.2ms) and is the fastest to reach the same accuracy (e.g., 0.99), which indicates that *by pruning redundant edges APG makes graph similarity search faster*. On the other hand, compared to the sparse counterparts, APG delivers better accuracy-latency tradeoff than both HNSW-sparse and NSG-sparse. APG is 34% and 66% faster than HNSW-sparse and NSG-sparse to reach the same accuracy (0.98) and achieves the highest accuracy under the same search time budget (e.g., <0.3ms), which indicates *it is preferable to first build a proximity graph with a large number of edges and then prune to obtain a simplified one than directly build a proximity graph with size comparable to the pruned graph.* Random pruning has the worst performance, because it considers neither the graph structure nor edge importance. For Deep1M and GloVe, APG outperforms HNSW-sparse, NSG-sparse, and HNSW-rand. Compared to HNSW and NSG, APG achieves almost on-par accuracy-latency tradeoff, but it has 50% less number of edges than HNSW and NSG (Fig. 5d) and therefore is much more memory efficient.

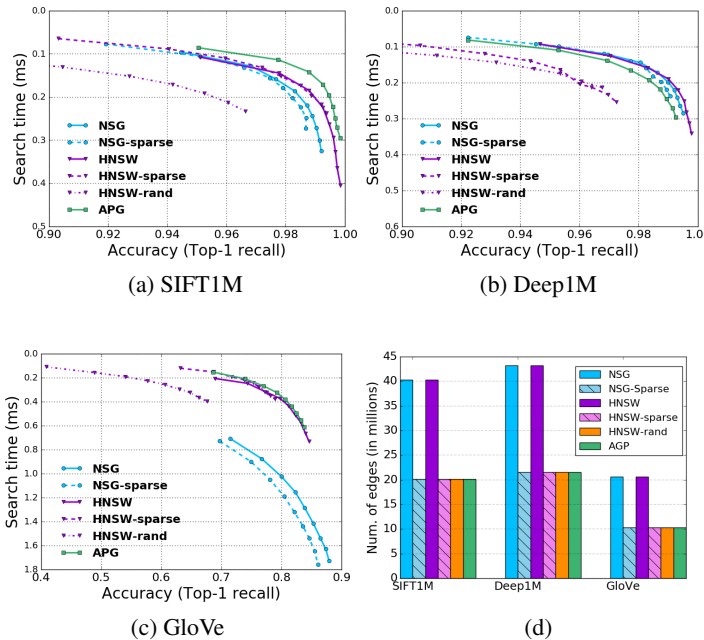

(a) SIFT1M

(b) Deep1M

(c) GloVe

(d)

Figure 5: Comparison to existing approaches and alternatives.

## 6 CONCLUDING REMARKS

Proximity graph is an important data structure for building large scale vector search engine of many machine learning systems. It is crucial to make it answer queries with low latency, low memory cost, and high accuracy. To the best of our knowledge, this is the first work on proximity graphs that demonstrate that we can learn to anneal and prune proximity graphs without losing much accuracy. This has several benefits: pruned edges reduce the memory cost; and the pruned proximity graphs perform similarity search 21–41% faster than existing and the state-of-the-art approaches with minimal loss of accuracy. The cost is a small investment on learning and optimization time. We open-sourced the code at https://drive.google.com/open?id=15vGhNS0O9l-zPAbdPAxwxIzV558jjGeQ.

---

[1]APG is built with $R = 64$ and a pruning ratio $\sigma = 0.5$. (1) and (2) are built with $R = 64$, (3) and (4) are built with $R = 32$. (5) is built with $R = 64$ but with 50% of edges randomly deleted.

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

## A    PROOF OF THEOREM

**Theorem A.1.** *Let $G_0(V, E_0)$ be the original graph to prune, and let $Q$ be the query set. Suppose there exists a subset of edges $E^* \subseteq E_0$ such that the average recall of retrieving the nearest neighbor in $V$ for all queries $q \in Q$ using edges in $E^*$ is 1. Let $G(k)$ be the random graph at iteration $k$ running Algorithm 1, and let $\{R_e(k)\}$ be the family of Bernoulli random variables defined on the edges of $G(k)$, with $\mathbb{P}(R_e(k) = 1) = p_e(k)$. Assume that $p_e(k_e) < \frac{1}{2}$ for $e \in E_0 \backslash E^*$ at some step $k_e$. Also assume that $T(k) \geq T(k+1)$ for all $k$. If $\sigma = 1 - |E^*|/|E_0|$, then:*

*(1)* $\sum_{e \in E^*} p_e(k + 1) \geq \sum_{e \in E^*} p_e(k)$ *for all sufficiently large $k$. In particular,* $\lim_{k \to \infty} \sum_{e \in E^*} p_e(k)$ *exists.*

*(2)* *If $\lim_{k \to \infty} T(k) = 0$, then $\lim_{k \to \infty} p_e(k) = 1$ for all $e \in E^*$, and $\lim_{k \to \infty} p_e(k) = 0$ for all $e \in E_0 \backslash E^*$.*

*(3)* *With the condition in (2), suppose $r(k)$ denotes the average expected recall of retrieving the ground truth nearest neighbor of a query $q \in Q$ in a subgraph of $G(k)$ randomly sampled from the joint distribution $\{R_e(k)\}$. Then $\lim_{k \to \infty} r(k) = 1$.*

*Theorem A.1.* We start with showing (1). Note that the probability sum constraints

$$\sum_{e \in E^*} p_e(k) + \sum_{e \in E_0 \backslash E^*} p_e(k) = (1 - \sigma)|E_0| = |E^*|$$

is satisfied for all $k$. It then suffices to show that

$$\sum_{e \in E \backslash E^*} p_e(k+1) \leq \sum_{e \in E \backslash E^*} p_e(k)$$

for sufficiently large $k$. Observe that $w_e(k)$ remains a constant ($= w_e(0)$) for all $k$ and all $e \in E_0 \backslash E^*$, and that $\mu(k)$ monotonically decreases as $k$ increases. We claim that $p_e(k+1) \leq p_e(k)$ for all $k \geq k_e$. Together with the probability sum constraints this shows that $\sum_{e \in E^*} p_e(k)$ is monotonically increasing for $k \geq \max_e k_e$. Since $\sum_{e \in E^*} p_e(k)$ is bounded from above, $\lim_{k \to \infty} \sum_{e \in E^*} p_e(k) = \sup_k \sum_{e \in E^*} p_e(k)$ exists.

For (2), since $\lim_k T_k = 0$, observe that $\lim_{k \to \infty} p_e(k) \to 0$ for all $e \in E_0 \backslash E^*$. Thus the probability sum constraint enforces that $\lim_k \sum_{e \in E^*} p_e(k) = |E^*|$. Recall that $\liminf_k x_k + \limsup_k y_k \geq \liminf_k (x_k + y_k)$ for any real sequences $x_k$ and $y_k$. Now consider any $e \in E^*$, we have

$$\liminf_{k \to \infty} p_e(k)$$
$$\geq \lim_{k \to \infty} \sum_{e \in E^*} p_e(k) - \limsup_{k \to \infty} \sum_{e' \in E^* \backslash \{e'\}} p_{e'}(k)$$
$$\geq |E^*| - \sum_{e' \in E^* \backslash \{e'\}} \limsup_{k \to \infty} p_{e'}(k)$$
$$\geq |E^*| - \sum_{e' \in E^* \backslash \{e'\}} 1$$
$$= |E^*| - (|E^*| - 1) = 1.$$

which implies that $\lim_k p_e(k) = 1$.

To prove (3), we first define the notation $\chi(q, E)$ to be the indicator function of whether the ground truth nearest neighbor of a query $q$ can be retrieved in $G$ using a subset of edges $E \subseteq E_0$. More precisely, $\chi(q, E) = 1$ if the nearest neighbor of $q$ can be retrieved using edges in $E$; and $\chi(q, E) = 0$ otherwise. Note that the average recall at step $k$ can be written as

$$r_k = \frac{1}{|Q|} \sum_{q \in Q} \sum_{E \subseteq E_0} \chi(q, E) \mathbb{P}(E \text{ is chosen from } G(k)),$$

where the second summation is amongst all subsets of $E_0$. Since each edge being chosen is independent of other edges, we can expand the probability and it follows that

$$r_k = \frac{1}{|Q|} \sum_{q \in Q} \sum_{E \subseteq E_0} \left( \chi(q, E) \prod_{e \in E} p_e(k) \prod_{e \notin E} (1 - p_e(k)) \right).$$

Since we assumed that $\chi(q, E^*) = 1$ for all $q \in Q$, it follows that $\chi(q, E) = 1$ for all $E \supseteq E^*$. Thus

$$r_k \geq \frac{1}{|Q|} \sum_{q \in Q} \sum_{E^* \subseteq E \subseteq E_0} \left( \prod_{e \in E} p_e(k) \prod_{e \notin E} (1 - p_e(k)) \right).$$

Note that

$$\sum_{E^* \subseteq E \subseteq E_0} \left( \prod_{e \in E} p_e(k) \prod_{e \notin E} (1 - p_e(k)) \right)$$

$$= \sum_{E^* \subseteq E \subseteq E_0} \left( \prod_{e \in E^*} p_e(k) \prod_{e \in E \setminus E^*} p_e(k) \prod_{e \notin E} (1 - p_e(k)) \right)$$

$$= \prod_{e \in E^*} p_e(k) \sum_{E^* \subseteq E \subseteq E_0} \left( \prod_{e \in E \setminus E^*} p_e(k) \prod_{e \notin E} (1 - p_e(k)) \right)$$

$$= \prod_{e \in E^*} p_e(k) \sum_{E^* \subseteq E \subseteq E_0} \mathbb{P}(E \text{ is chosen from } E_0 \setminus E^*)$$

$$= \prod_{e \in E^*} p_e(k).$$

Therefore

$$\liminf_{k \to \infty} r_k \geq \liminf_k \frac{1}{|Q|} \sum_{q \in Q} \prod_{e \in E^*} p_e(k)$$

$$\geq \frac{1}{|Q|} \sum_{q \in Q} \liminf_{k \to \infty} \prod_{e \in E^*} p_e(k)$$

$$\geq \frac{1}{|Q|} \sum_{q \in Q} \prod_{e \in E^*} \liminf_{k \to \infty} p_e(k)$$

$$= \frac{1}{|Q|} \sum_{q \in Q} 1 = 1.$$

This shows that $\lim_{k \to \infty} r_k = 1$ and completes the proof of the Theorem. $\square$

## B    FREQUENCY DISTRIBUTION OF DEGREE

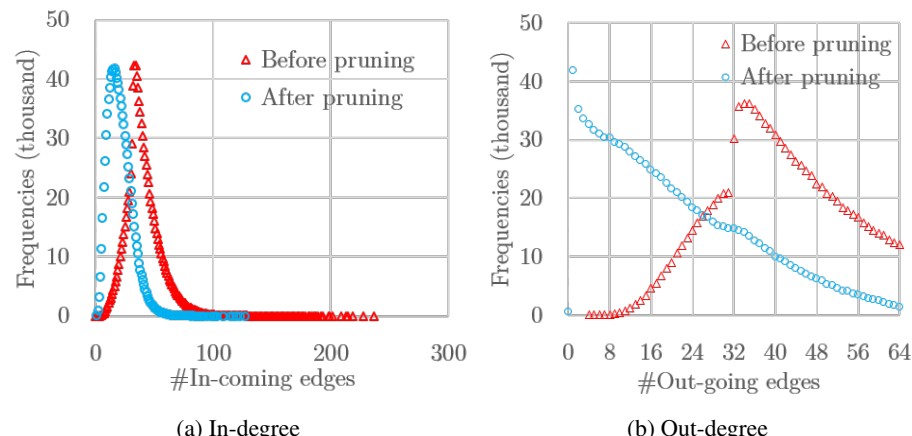

(a) In-degree                                         (b) Out-degree

Figure 6: Frequency distributions of in-degree and out-degree of proximity graph before and after pruning over SIFT1M.

## C    INTERNAL OF APG LEARNING PROCESS

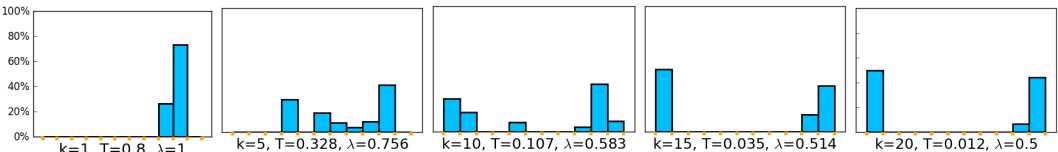

Figure 7: Distribution of keep probability at different steps of the optimization.

