# OpenReview forum: "Learning to Anneal and Prune Proximity Graphs for Similarity Search"
_ICLR.cc/2020/Conference — Reject_

### Official Review · AnonReviewer2 · 2019-10-16
**Official Blind Review #2**

**Rating:** 8

**Review:**


Summary:
The authors propose an extension of proximity graphs, Annealable Proximity Graphs (APG) for similarity search. APG augments pre-built proximity graph with a learnable weight and a keep probability with each edge in the graph which are updated in an iterative manner. This allows identifying important edges while preserving graph monotonicity which is important for search efficiency. All the edge weights are initialized with uniform weights and are updated using a stochastic optimization algorithm. Once the optimization finishes, edge weights are ranked and less important edges are pruned as per the desired pruning ratio. The authors also theoretically prove convergence and correctness of their proposed algorithm. The results demonstrate that APG maintains almost the same accuracy while decreasing search time by 21-41 %. Overall, I find that the proposed method and its results are convincing. The paper is very well written and all steps are rightly justified.

Questions:
1. In Figure 4, for all three datasets, the performance starts to drop exactly after 50%. Can you provide any intuition behind this consistent pattern across all datasets?

2.  In Figure 7(b), can you explain the cause behind the discontinuity of the “Before Pruning” plot?


**Experience Assessment:**

I do not know much about this area.

**Review Assessment: Checking Correctness Of Derivations And Theory:**

I assessed the sensibility of the derivations and theory.

**Review Assessment: Checking Correctness Of Experiments:**

I assessed the sensibility of the experiments.

**Review Assessment: Thoroughness In Paper Reading:**

I read the paper at least twice and used my best judgement in assessing the paper.

---

> ### Author Response · Authors · 2019-11-13
> **Thank you for your review!**
>
> Thanks for the comments! We really appreciate it. Please find our response below and let us know if you have any further questions.
>
> Comment: In Figure 4, for all three datasets, the performance starts to drop exactly after 50%. Can you provide any intuition behind this consistent pattern across all datasets?
>
> Response: The performance starts to drop after 50% because, in that particular experiment, we set the sampling rate λ(K) (in Equation 4) to 0.5 during the iterative optimization, which means the optimization constantly force half of the edges to be useful for preserving the graph monotonicity. As a result, when pruning more than 50% of edges, the performance starts to drop considerably. If the goal is to get the best accuracy, then besides the pruning rate σ, the sampling rate in the optimization process also needs to be adjusted.
>
> Comment: In Figure 7(b), can you explain the cause behind the discontinuity of the “Before Pruning” plot?
>
> Response: This jump at 32 means there are around 20K nodes that have an out-going degree of 32. Note that this distribution is from the baseline HNSW. HNSW is a hierarchical graph, where upper layers are recursively sampled from the bottom layer. While nodes at the bottom layer have maximally R = 64 outgoing edges for each node, nodes at the upper layers (used for fast routing) have 32 (R/2) outgoing edges per node. So those nodes with 32 edges are contributed by those upper layers.

---

### Official Review · AnonReviewer3 · 2019-10-22
**Official Blind Review #3**

**Rating:** 6

**Review:**

This paper suggests an approach for learning how to sparsify similarity search graphs. Graph-based methods currently attain state of the art performance for similarity search, and reducing their number of edges may speed them up even further. The paper suggests a learning framework that uses sample queries in order to determine which edges are more useful for searches, and prune the less useful edges. This is a sensible and potentially useful approach in line with the recent flurry of work on improving algorithms with tool from machine learning.

While I like the overall approach and believe it could work, the experiments seem to have some weaknesses:

1. It is not clear to me why Table 1 contains only UPG and APG with pruning half the edges, without natural pruning baselines like uniformly subsampling the edges by a factor of half, or constructing the graph with half as many edges to begin with. Both of these baseline appear in the plots afterwards, which suggest very similar performance to APG, and it would be interesting to see the numbers side by side. The numbers for UPG and APG alone do not say much: the fact that the number of edges drops by half and the search speed drops by somewhat less than half are inevitable artifacts of the construction. The interesting part is the effect on the accuracy, and its quality is hard to assess without comparison to any baselines.

2. The plots leave the impression that the proposed algorithm does not actually perform that well. It is superior on SIFT, but does not improve performance on GloVe, and is outperformed on Deep1M. This seems to render the textual description of the results somewhat overstated, if I read it right (is it referring only to SIFT?).

In conclusion, while I am optimistic about the paper and the approach, I am tentatively setting my score below the bar in light of the somewhat unsatisfactory experimental performance. The paper would be significantly helped by showing non-negligible improvement on more than one dataset or in more settings.

Other comments:
1. What is NSG? I could not find a spelling out of the abbreviation nor a reference.
2. HNSW-sparse and HNSW-rand are very nearly impossible to tell apart in the plots. I suggest using a clearer visual distinction.
3. "Interestingly, pruning provides the benefits of improved search efficiency" - isn't that the point of pruning?
4. It is curious that using HNSW with R=32 instead of R=64 hurts the performance so much on Deep1M, while it has hardly any effect on SIFT and GloVe, do you perhaps have an explanation for this result?


**Experience Assessment:**

I have read many papers in this area.

**Review Assessment: Checking Correctness Of Derivations And Theory:**

I assessed the sensibility of the derivations and theory.

**Review Assessment: Checking Correctness Of Experiments:**

I assessed the sensibility of the experiments.

**Review Assessment: Thoroughness In Paper Reading:**

I read the paper at least twice and used my best judgement in assessing the paper.

---

> ### Author Response · Authors · 2019-11-13
> **On performance improvements over existing approaches**
>
> Thank you for taking the time to review our paper and provide suggestions! We try to answer your questions in order:
>
> Comments: Both of these baselines appear in the plots afterward, it would be interesting to see the numbers side by side.
>
> Response:  We added the side-by-side comparison to the other two baselines in Table 1. To compare the performance of baselines and our approach apples to apples, we performed controlled experiments to keep all approaches to reach the same accuracy target in order to compare the edge reduction rate and the search time improvement.  The results are shown below (also in Table 1 of the revised submission).
>
> Dataset	| Configuration	| Recall@1	| #Edges | Latency(ms)	| Edge reduction	| Search time improvement
> SIFT1M	| Original PG	| 0.993	        | 40.3M	  | 0.23	                | 50.0%	                | 26.1%
>  	        | PG + sampling	| 0.992	        | 20.2M	  | 0.43	                | 0.0%	                |60.5%
>  	        | Sparse-PG	        | 0.991	        | 20.2M	  | 0.66	                | 0.0%	                |74.2%
>   	        | Ours	                | 0.993	        | 20.2M	  | 0.17
> Deep1M	| Original PG	| 0.947	        | 43.2M	  | 0.1	                | 50.0%	                |-4.0%
>  	        | PG + sampling	| 0.945	        | 21.6M	  | 0.16	                | 0.0%	                |35.0%
>  	        | Sparse-PG	        | 0.943	        |21.6M	  | 0.14	                | 0.0%	                |25.7%
>  	        | Ours	                | 0.943	        |21.6M 	  | 0.104
> GloVe1M|Original PG	        | 0.831	        |20.6M	  | 0.55	                | 50.0%	                |1.8%
>  	        |PG + sampling	| 0.814	        |10.3M	  | 1.45	                | 0.0%	                |62.8%
>  	        |Sparse-PG	        | 0.83	        |10.3M	  | 0.74	                | 0.0%	                |27.0%
>  	        |Ours	                |0.833	        |10.3M	  | 0.54
>
> Overall, compared to the original proximity graph,  our approach reduces the number of edges by  50%, making  PG more memory efficient. On SIFT1M and GloVe1M, APG further makes the search 26.1% and 1.8% faster, respectively. Both PG + sampling and Sparse-PG have a similar number of edges like ours. However, both have worse search efficiency, and our approach is overall 25--74.2% faster compared to these two approaches. As for why heuristic pruning has different search efficiency, please refer to our answer to your question 3.
>
>  Other comments:
>
> 1. What is NSG? I could not find a spelling out of the abbreviation nor a reference.
>
> Response: NSG refers to Navigable Spread-out Graph in [Fu et al 2019]. We have added the abbreviation in the paper.
>
> 2. HNSW-sparse and HNSW-rand are very nearly impossible to tell apart in the plots. I suggest using a clearer visual distinction.
>
> Response: Thanks for the suggestion. We have tried to fix it in the revised submission.
>
> 3. "Interestingly, pruning provides the benefits of improved search efficiency" - isn't that the point of pruning?
>
> Response: Yes and no. The answer is perhaps less intuitive than it sounds. If the search takes a fixed number of hops to reach the closest vector, then pruning edges indeed will lead to less number of vectors being checked. However, in the graph-based algorithm, the stop condition is (1) doing greedy search until reaching a local optimum and (2) search budget has been exhausted, so the search may end up searching the same amount of edges even after pruning. Alternatives such as subsampling and choosing a sparse configuration to start with hurt the graph properties, which proximity graphs rely on for search efficiency. Without those properties, a query can take a detour or get stuck at a local optimum more easily. To the best of our knowledge, prior to our work, it was not clear whether the destructive pruning process would affect search efficiency for proximity graphs.
>
> Apart from the search efficiency, the goal of pruning is to also simplify the proximity graph by reducing the number of redundant edges and so to make it more memory efficient for online search scenarios.
>
> 4. It is curious that using HNSW with R=32 instead of R=64 hurts the performance so much on Deep1M, while it has hardly any effect on SIFT and GloVe, do you perhaps have an explanation for this result?
>
> Response: Yes, HNSW, and in general many other proximity graphs,  has connectivity issues when the number of out-going edges (R) is small. That is, it can have "isolated nodes or subgraphs" --- nodes with zero in-degree or weakly connected subgraphs that are unreachable. This issue is data-dependent. One can imagine that in regions where the density of nodes is high, the graph should have more edges to capture the closeness relationship among nodes. When R is not large, some of that closeness relationship is going to be missing. For Deep1M, it is possible that its vector distribution is more skewed so that reducing out-going edges hurts the performance more than SIFT and GloVe.

---

> > ### Comment · AnonReviewer3 · 2019-11-15
> > **I have read the author response**
> >
> > I thank the authors for their response and clarifications. I believe the main point in my review stands: the paper is able to exhibit improved running time on only one of the three datasets, whereas on the other it only slightly improves or slightly degrades the running time. Reducing space by half by decreasing the number of edges is an advantage, but I am not sure it is by itself a sufficient justification for the method, if it does not consistently improve the running time. I will reconsider my position on the paper during the discussion phase.

---

### Official Review · AnonReviewer1 · 2019-10-28
**Official Blind Review #1**

**Rating:** 6

**Review:**

This paper studies the problem of improving proximity graph for nearest neighbor search. It formulates the task of pruning the graph as a problem of learning annealable proximity graph. A hard pruning processes is used after the learning process, and the results shows that the proposed method can reduce 50% of the edges and speed up the search time by 16-41%.

The biggest concern I have is how to evaluate the performance.  The proposed method is mainly based on the comparison with [Malkov 2016], which did not use an extra training set to learn the NPG as proposed in this paper. So it is not surprising the proposed method will perform better. I would like to see more comparisons with at least the following methods: (1) a heuristic pruning strategy (2) the state of the arts of tree based NN search and hashing based search (3) the recent work in proximity graph [Fu et al 2019]

To summarize, I think the paper studies an important problem and the proposed method is reasonable. However, I cannot be convinced it is the state of the art for large scale nearest search unless I see more comparisons in the new version.

Detailed comment:
- in section 5.2, "APG reduce the number of edges by 2 times " -> "APG reduce the number of edges by 50\%"

**Experience Assessment:**

I have read many papers in this area.

**Review Assessment: Checking Correctness Of Derivations And Theory:**

I assessed the sensibility of the derivations and theory.

**Review Assessment: Checking Correctness Of Experiments:**

I did not assess the experiments.

**Review Assessment: Thoroughness In Paper Reading:**

I made a quick assessment of this paper.

---

> ### Author Response · Authors · 2019-11-13
> **On experimental comparisons to existing work**
>
> We’re thankful to the reviewer for the comments and suggestions. The reviewer's main concern was on the performance evaluation and suggested several comparisons, which we agree are all very important points and we address below one-by-one. Please let us know if you have further questions.
>
> Comment: Comparison to  the recent work of proximity graph in [Fu et al 2019]
>
> Response: While doing the performance evaluation,  we actually compared with [Fu et al 2019] in section 5.3, which we referred to as NSG (Navigable Spread-Out graph) as the name used in the original paper. Our approach achieves better latency--accuracy tradeoff than NSG and a heuristic approach to prune NSG (NSG-sparse). As an example, on SIFT1M, to reach the same target accuracy (e.g., 99%),  NSG takes 0.27ms, while our approach takes 0.17ms, which is 37% faster than NSG. Similarly, when given the same search budget (e.g., 0.3ms), NSG achieves 0.9% error rate, while our approach achieves 0.16% error rate, which is a relative 82% error rate reduction over NSG.  And we would like to point out that although we demonstrate the improvement through pruning the HNSW graph, our approach is completely complementary to NSG and could potentially increase its performance as well, since fundamentally both HNSW ([Malkov 2016]) and NSG ([Fu et al 2019]) are proximity graphs that approximate the Delaunay graph, and the graph properties they show are very similar.
>
> Comments: Comparison to a heuristic pruning strategy
> Response: We actually compared with two heuristic pruning strategies in section 5.3: (1) pruning by uniformly subsampling the edges, and (2) directly restricted the number of edges of each node, which effectively pruned the edges during the graph construction phase, and we evaluated the heuristic approach on three methods: EFANNA, HNSW, and NSG (called EFANNA-sparse, HNSW-sparse, NSG-sparse, respectively).  Random pruning leads to worse search efficiency either because a query needs to take a detour to find the nearest neighbor or it may not even find the nearest neighbor if the graph becomes disconnected.  Direct pruning leads to worse performance because it puts a small cap on nodes that need an intrinsically large number of edges to preserve graph monotonicity, and search can easily get stuck at a local minimum (as described in the challenge section).  We have revised the paper to put these comparisons side-by-side in Table 1 so that they are easier to see.
>
> Comments: Comparison to the state of the arts of tree-based NN search and hashing based search
> Response:  It is fairly easy for us to add these results. However, several prior works have studied and reported the comparison results between graph-based approaches and other approaches including tree-based and hashing based methods  ([Fu et. al., 2019], [Aumüller et. al.], [Douze et. al.]). Overall, proximity graphs outperform other approaches by a large margin. Looking at comparisons in https://github.com/erikbern/ann-benchmarks ([Aumüller 2018]), the proximity graphs are the state-of-the-art methods for similarity search in regards to search time and accuracy.  So we focused our comparison on graph-based approaches.
>
> More specifically, Tree structure-based algorithms, and in general space-partitioning based approaches are known to work well in low dimensions but becomes really challenging to partition the subspaces, especially in high dimensional space, so that neighbor areas can be scanned efficiently to identify the nearest neighbors of a given query. The complexity of these approaches is O(D x N^(1 - 1/D), which gradually becomes not more efficient than a brute-force search as the dimension D becomes large (e.g., >32).  Hashing based methods, such as Locality-Sensitive Hashing (LSH), have theoretically derived guarantees of worst-case search complexity. However, LSH and similar approaches have been designed for large bag-of-words sparse vectors with hundreds of thousands of dimensions, not dense continuous vectors with only a few hundreds of dimensions, like those learned by neural networks. Prior work has demonstrated that proximity graph-based approaches outperform hashing-based approaches by a large margin on large-scale datasets.
>
> We hope these clarifications give more evidence on your concern raised on the “experimental comparisons to existing work”. As it was your main concern in your review, we would like to ask you if you could reconsider your score?
>
> "Fast Approximate Nearest Neighbor Search With The Navigating Spreading-out Graph", Fu et. al. VLDB 2019. https://arxiv.org/abs/1707.00143
> "ANN-Benchmarks: A Benchmarking Tool for Approximate Nearest Neighbor Algorithms", Aumüller et. al., 2018. https://arxiv.org/pdf/1807.05614.pdf
> "Link and code: Fast indexing with graphs and compact regression codes", Douze et. al., CVPR 2018, https://arxiv.org/pdf/1804.09996.pdf

---

### Decision · Program_Chairs · 2019-12-19

**Decision:**

Reject

**Comment:**

The paper proposes a method to prune edges in proximity graphs for faster similarity search. The method works by making the graph edges annealable and optimizing over the weights. The paper tackles an important and practically relevant problem as also acknowledged by the reviewers. However there are some concerns about empirical results, in particular about missing comparisons with tree-structure based algorithms (perhaps with product quantization for high dimensional data), and about modest empirical improvement on two of the three datasets used in the paper, which leaves room for convincing empirical justification of the method. Authors are encouraged to take the reviewers' comments into account and resubmit to a future venue.